# Effectiveness of Nursing Interventions on Preventing the Risk of Infection in Adult Inpatients: Protocol for a Systematic Review

**DOI:** 10.3390/nursrep15060210

**Published:** 2025-06-11

**Authors:** Luís Filipe Pereira Todo Bom, Ema Soraia Fazenda Mata, Helena Margarida Pereira Cunha, Maria do Céu Mendes Pinto Marquês, Maria dos Anjos Dixe

**Affiliations:** 1Comprehensive Health Research Centre (CHRC), University of Évora, 7000-801 Évora, Portugal; mcmarques@uevora.pt; 2School of Health Sciences, Polytechnic University of Leiria, Campus 2, Morro do Lena, Alto do Vieiro, Apartado 4137, 2411-901 Leiria, Portugal; ema.mata@ipleiria.pt (E.S.F.M.); helenacunha04@gmail.com (H.M.P.C.); maria.dixe@ipleiria.pt (M.d.A.D.); 3Centre for Innovative Care and Health Technology (ciTechCare), Polytechnic University of Leiria, Campus 5, Rua das Olhalvas, 2414-016 Leiria, Portugal; 4Local Health Unit of the Region of Leiria, Rua de Santo André, 2410-197 Leiria, Portugal; 5Department of Nursing, University of Évora, 7000-811 Évora, Portugal

**Keywords:** healthcare-associated infections, nursing interventions, infection prevention, patient safety, systematic review

## Abstract

**Background/Objectives:** Healthcare-associated infections (HAIs) are a major global public health concern, significantly impacting patient safety and healthcare quality. These infections are associated with high morbidity and mortality rates, prolonged hospital stays, and increased healthcare costs. Nurses play a critical role in infection prevention, implementing evidence-based interventions to reduce infection risks. This systematic review aims to identify and synthesize the most effective nursing interventions to prevent HAIs in hospitalized adults, analyzing their variability across different clinical settings and populations. **Methods:** This systematic review follows the Joanna Briggs Institute (JBI) methodology for systematic reviews of effectiveness and is reported according to PRISMA guidelines. The protocol is registered in the PROSPERO database (CRD42024582820). This review includes randomized controlled trials, quasi-experimental studies, and observational studies (cohort, case-control, and cross-sectional) assessing the effectiveness of nursing interventions in reducing HAIs. A comprehensive search is conducted in the PubMed, CINAHL, Scopus, Web of Science, and Cochrane databases. Study selection, data extraction, and quality assessment are performed by two independent reviewers, with disagreements resolved by a third reviewer. **Results:** The primary outcomes include reductions in HAI incidence rate, increased adherence to preventive interventions, decreased hospital length of stay, reduced readmission rates due to infections, and overall patient safety improvements. A meta-analysis is conducted when feasible; otherwise, results are synthesized narratively. **Conclusions:** The findings of this review contribute to the standardization of evidence-based nursing practices for HAI prevention, promoting safer healthcare environments. By identifying the most effective interventions, this study aims to support healthcare professionals and policymakers in implementing targeted infection control strategies.

## 1. Introduction

Healthcare-associated infections (HAIs) represent one of the greatest challenges to patient safety and quality in healthcare services and are a serious public health problem. These infections, which are often acquired during hospitalisation, are associated with high rates of morbidity and mortality, as well as placing an economic burden on health systems worldwide. According to the World Health Organisation (WHO), millions of hospitalised people are affected by HAIs every year, resulting in clinical complications, longer hospital stays, and an increase in the workload of healthcare professionals. In developed countries, it is estimated that between 5% and 10% of hospitalised patients acquire at least one HCAI, while in developing countries this rate can exceed 10% [1].

Preventing these infections is an essential pillar of patient safety and requires the implementation of measures based on scientific evidence. Strategies such as hand hygiene, proper use of barriers, rigorous disinfection of hospital surfaces, and specific protocols for handling invasive medical devices are widely recommended by health regulatory agencies [2]. However, adherence to these practices still shows significant variations between hospital units and countries, influenced by factors such as infrastructure, the hands of health professionals [3], workload, and organisational culture [4].

HAIs have been a global concern since the 19th century, with Florence Nightingale, Ignaz Semmelweis, and Joseph Lister pioneering the implementation of preventative measures [5]. Today, their relevance continues to grow due to their direct impact on population and health systems, resulting in millions of deaths every year and high costs [6]. Health professionals play a crucial role in preventing these infections, requiring a coordinated and effective response from health systems [7]. The approach to mitigate the risk of infection must be integrated and multidisciplinary, and it should include new technologies as a means of strengthening the control of hospital infections [8,9].

The implementation of case bundles, a set of standardised measures, has shown a significant reduction in HAIs [10]. Hand hygiene remains one of the fundamental pillars in the prevention of HCAI [11], with studies indicating reductions of up to 40 per cent in hospital-acquired infections in institutions that have implemented strict protocols [12]. In addition, environmental disinfection, particularly with technologies such as ultraviolet light, has been associated with a reduction of up to 30 per cent in infection rates [13].

Given the relevance of the risk of infection and the need for effective strategies to prevent it, it is essential to carry out a systematic review that consolidates the available evidence on the effectiveness of nursing interventions in reducing the risk of infection in hospitalised adults. Nursing plays a central role in implementing these preventive measures, as nurses are primarily responsible for the direct care of adult inpatients and for applying infection control protocols. Ongoing training for these professionals is essential to ensure adherence to best practices, with studies showing that regular training programmes significantly reduce hospital infection rates [14].

Antimicrobial resistance is a global challenge that requires effective strategies, where nurses play an essential role in its prevention. The One Health [15] initiative highlights the importance of education, monitoring antimicrobial use, and implementing stewardship programmes. However, the scarcity of specific training and the variability in adherence to evidence-based practices limit the effectiveness of interventions. Strengthening the role of nurses in preventing antimicrobial resistance is essential to improve patients’ safety and clinical outcomes [16].

This systematic review aims to systematise the most effective nursing interventions for preventing the risk of infection in adult inpatients, analysing their variability in different clinical settings and types of hospitalised population. It also seeks to compare prevention practices and policies across different cultural and geographical settings, identifying successful evidence-based strategies. The findings of this review could contribute to the standardisation of effective guidelines, promoting significant improvements in patient safety and the quality of care provided. Understanding the variability of interventions and identifying those with the greatest impact will allow for the formulation of more tailored and effective recommendations, benefiting both patients and healthcare professionals

However, despite the availability of evidence-based interventions to prevent healthcare-associated infections (HAIs), their effectiveness in real-world settings is often compromised by inconsistent adherence among healthcare professionals, particularly nurses. Studies have shown that even well-established practices such as hand hygiene or the use of care bundles are not uniformly implemented across institutions. Variability in adherence can be influenced by multiple factors, including workload, organizational culture, perceived usefulness of the interventions, and lack of continuous training. Therefore, understanding and addressing the barriers to nurses’ adherence is essential for maximizing the impact of infection prevention strategies and ensuring patient safety [4]. 

## 2. Materials and Methods

This proposed systematic review was conducted in accordance with the Joanna Briggs Institute (JBI) Guidelines for Systematic Reviews of Effectiveness [17] and reported in accordance with the Preferred Reporting Items for Systematic Reviews, Review Guidelines, and Meta-Analyses (PRISMA) [18]. This protocol is registered and publicly available in the PROSPERO database (CRD42024582820). This study does not require ethical approval, as the systematic review methodology involves reviewing and collecting data from publicly available materials.

### 2.1. Review Questions

The review question was developed to reflect the general and specific objectives of this systematic review, ensuring the comprehensive collection and synthesis of data on effective interventions to prevent the risk of infection:

‘What are the most effective nursing interventions to prevent the risk of infection in adult inpatients and how do these interventions vary between different clinical settings, populations, and geographical regions?’

### 2.2. Identifying Relevant Studies

This review was conducted using the PICOS strategy (Population, Intervention, Comparison, Outcomes, and Study Design). The initial selection criteria are defined in accordance with this approach.

#### 2.2.1. Participants

Studies involving adults (≥18 years) hospitalised in healthcare services are included, regardless of the length of stay and the pathology that led to their hospitalisation. Studies involving paediatric populations, neonates, or people admitted to intensive care units are excluded.

#### 2.2.2. Interventions

The interventions analysed include nursing interventions to prevent HCAI, such as hand hygiene, proper use of personal protective equipment (PPE), management of patients in hospital isolation, bundles for the prevention of infections associated with invasive medical devices, health education and continuing training for professionals, patients and the community, monitoring and auditing of practices, disinfection of medical devices, environmental hygiene and waste management, promotion of safe care, education on antimicrobial resistance, and the use of innovative technologies. Studies that are not conducted by nurses, that are not directly related to the control and risk of hospital-acquired infections, or that focus exclusively on the prescription of antimicrobials without involving nursing practices are excluded.

#### 2.2.3. Comparators

Comparisons are be made between different clinical settings, types of hospitalised populations, infection risk prevention practices, and policies in different cultural and geographical contexts. Studies evaluating care delivery models and their impact on patient outcomes are of interest.

#### 2.2.4. Outcomes

The outcomes analysed include a reduction in the incidence rate of healthcare-associated infections, an increase in the rate of adherence to preventive interventions (hand hygiene, use of sterile barriers, and compliance with protocols), a reduction in mortality and morbidity associated with hospital-acquired infections, a reduction in hospital stay length, a reduction in rate of re-hospitalisation due to complications, a reduction in costs associated with infections treatments, and an improvement inpatient’s satisfaction and perception of the quality of care received.

#### 2.2.5. Study Selection

The types of study considered include randomised clinical trials, quasi-experimental studies, experimental studies, and observational studies (cohort, case-control, and cross-sectional). Studies without adequate control, with poor methodological quality or that present incomplete or unavailable data, or whose full-text version is not accessible, even after contacting the authors, are excluded.

There are no restrictions on the period of publication, allowing the inclusion of all available evidence. However, only studies published in English, Portuguese, and Spanish are included.

### 2.3. Selecting Studies for Analysis

The search strategy is conducted with the aim of identifying both published and unpublished studies, ensuring a comprehensive review of the available literature on the subject. To this end, a preliminary search is carried out in the PubMed database in order to identify relevant articles, the titles and abstracts of which are analysed to develop a detailed search strategy. This strategy is applied to different databases, adapting the keywords and index terms according to the specificity of each database. The databases that are searched include PubMed, CINAHL (EBSCOhost), Scopus, Web of Science, Cochrane Central Register of Controlled Trials, Cochrane Database of Systematic Reviews, and Cochrane Methodology Register and Cochrane Clinical Answers (EBSCOhost), ensuring that the main sources of scientific evidence are included (Appendix A). In addition, the reference lists of selected studies are scrutinised to identify additional articles that may meet the inclusion criteria. Sources of grey literature are also consulted, covering unpublished studies and institutional documents relevant to the topic.

### 2.4. Study Selection

After the search, all the citations identified are collected and uploaded to the Rayyan Intelligent Systematic Review (Qatar Computing Research Institute, Doha, Qatar), where duplicates are removed. A pilot test is then carried out with 10 studies to ensure clarity and consistency in the application of the inclusion and exclusion criteria during the screening of titles and abstracts. The screening of titles and abstracts is conducted by two independent reviewers, based on the eligibility criteria. Studies considered potentially relevant are retrieved in full and their citation details are imported into the JBI System for the Unified Management, Assessment, and Review of Information (JBI SUMARI; JBI, Adelaide, Australia) [19]. Two independent reviewers assess the full text of the selected citations based on the inclusion criteria, following the PRISMA guidelines. The reasons for excluding studies that do not fulfil the inclusion criteria are documented and presented in this systematic review. Any disagreements between the reviewers during the selection process are resolved through discussion or with the intervention of a third reviewer. The search results, study selection process, and inclusion criteria are fully documented in this final systematic review and presented in a PRISMA flow chart [20].

### 2.5. Assessment of Methodological Quality

Eligible studies are critically assessed for methodological quality in this review. This assessment is carried out by two independent reviewers, using the JBI critical appraisal tools for each study methodology [17] (for example, the checklist for quasi-experimental studies or randomised clinical trials). The authors of the articles are contacted to request missing data or additional information when necessary. Any disagreements are resolved through discussion or with the intervention of a third reviewer. All studies, regardless of the results of the methodological quality assessment, are included in the data extraction and synthesis process. The results of the critical appraisal are presented in a table accompanied by a narrative analysis.

### 2.6. Data Extraction

Data extraction from studies included in the review is carried out by two independent reviewers using the standardised JBI data extraction form. The data extracted include specific details about participants, the intervention (time of application, frequency, and duration), the comparison group, and the findings before, during, and after the intervention that are relevant to the review question, as well as the study design (Appendix B). In the event of disagreement between the reviewers, resolution is reached through discussion or with the consultation of a third reviewer.

### 2.7. Data Synthesis

Data are summarised, taking into account the availability and nature of the evidence extracted from the selected studies. Whenever possible, quantitative data are analysed using meta-analysis [21], while qualitative data are presented by means of a narrative synthesis. Various results are evaluated, including the incidence rate of healthcare-associated infections (HAIs), considering their frequency in different hospital contexts and the factors associated with their variation. Healthcare professionals’ adherence to hygiene and infection prevention protocols, such as hand hygiene, the use of barriers, and compliance with standard precautions, is examined to determine their effectiveness in reducing the transmission of infectious agents. In addition, the impact of effective interventions on reducing the risk of infection is analysed, as well as the influence of these measures on reducing the length of hospital stay and readmission rates. The economic analysis includes assessing the reduction in costs associated with treating hospital-acquired infections, comparing different prevention and control strategies. Environmental control is evaluated, considering the effectiveness of waste management measures in reducing the risk of infection. Inpatient satisfaction and perception of the quality of care received are also be analysed in an attempt to identify the impact of interventions on inpatient’s experiences. Statistical analyses are conducted using SPSS Version 29.0.2.0 (20) and Meta-Essentials software for Microsoft Excel Version 16.97.2. In the event that meta-analysis is not feasible due to insufficient data or high heterogeneity, the findings are presented in a narrative synthesis, including tables and figures, to facilitate the interpretation of the results. This way, the results are organized in a clear and objective manner, ensuring a comprehensive and rigorous interpretation of the findings of this systematic review.

### 2.8. Assessing Certainty in the Findings

The Grading of Recommendations, Assessment, Development, and Evaluation (GRADE) approach is followed to grade the certainty of the evidence, regardless of the synthesis approach used (whether meta-analysis or narrative synthesis). The GRADE framework assesses the strength of evidence from randomised clinical trials (RCTs) and non-randomised studies. The data from each type of study are analysed separately, allowing informed judgements to be made about the quality of the evidence and the level of confidence in the outcomes for each type of study. The outcomes are presented in a Summary of Findings (SoF) created using GRADEpro GDT (McMaster University, Hamilton, ON, Canada). The SoF presents, where applicable, the following information: absolute risks for the treatment and control groups, estimates of the relative risk, and a classification of the quality of the evidence, based on the risk of bias, directionality, heterogeneity, precision, and risk of publication bias of the review results.

## 3. Results

This review began in January 2025 and the official literature search was completed by the end of March 2025. The results are presented in three main formats: summary tables, visual figures, and a narrative synthesis. A PRISMA flow chart for the study selection process is presented in advance. A conceptual map was developed to illustrate the relationships between the main themes identified in the review, providing an overview of the results. The dissemination strategy involves publishing the results of this review in an open-access, peer-reviewed health journal and presenting the results at prestigious scientific conferences. The results will be disseminated to an academic audience by June 2025. In addition to the academic audience, the dissemination strategy will also target frontline healthcare professionals, infection prevention and control teams, healthcare managers, and policymakers. To ensure accessibility and encourage the translation of evidence into practice, we intend to develop technical briefs, executive summaries, and visual materials such as infographics which will be made available through institutional channels and professional networks. This broader dissemination approach aims to maximise the practical impact of the review’s findings across multiple levels of healthcare delivery.

## 4. Limitations and Strengths

This systematic review presents several strengths. Firstly, it is grounded in a rigorous methodological framework provided by the Joanna Briggs Institute (JBI) for reviews of effectiveness, ensuring transparency, reproducibility, and a robust appraisal of the included studies. The review protocol is registered in the PROSPERO database (CRD42024582820) and its reporting follows the PRISMA 2020 guidelines, increasing methodological integrity. The search strategy is comprehensive and includes major databases, grey literature, and reference lists, with no time restrictions, capturing the widest possible range of relevant evidence. Nevertheless, some limitations must be acknowledged. One potential limitation concerns the inclusion of only studies published in English, Portuguese, and Spanish. This linguistic restriction might lead to the exclusion of potentially relevant studies published in other languages, introducing a risk of language bias. In addition, the inclusion of different study designs (RCTs and quasi-experimental and observational studies) may lead to heterogeneity, limiting the possibility of conducting a meta-analysis. In such cases, a narrative synthesis will be used, which, despite being rigorous, lacks the statistical power of meta-analytical techniques. Another significant limitation relates to the variability in definitions, outcome measurements, and intervention protocols across studies. Infection prevention practices can differ considerably across healthcare systems and geographical regions, affecting the comparability and generalizability of results. Differences in healthcare infrastructure, staff–patient ratios, and infection control policies may influence the implementation and effectiveness of interventions. Moreover, potential publication bias remains a concern, as studies with positive outcomes are more likely to be published. Although grey literature is consulted, it is not always accessible or well-indexed, which could hinder a full representation of existing evidence. Finally, the effectiveness of interventions may be affected by contextual factors such as organizational culture or staffing constraints, which are not always detailed in the primary studies. To mitigate these limitations, rigorous methodological quality appraisal using JBI tools and the GRADE approach is employed. The findings are interpreted cautiously and limitations are explicitly addressed in the final analysis.

## 5. Conclusions and Implications for Practice

It is hoped that this systematic review will make a relevant contribution to nursing science and practice by identifying and synthesising the most effective nursing interventions for preventing healthcare-associated infections (HAIs) in hospitalised adults. The objective outlined in this protocol guides the methodological structure of the review, taking into account the variability of clinical contexts and populations. The expected results should support the standardisation of clinical practices based on the best available evidence, facilitating the development of guidelines adapted to different healthcare realities; guide the decision-making of nurses, managers, and trainers, allowing them to prioritise infection control strategies with the greatest clinical and economic impact; strengthen the fight against antimicrobial resistance by highlighting preventive interventions that reduce the unnecessary use of antibiotics and, consequently, selective pressure; and identify gaps and opportunities for innovation, namely the incorporation of digital tools, automatic learning methods for infection surveillance, and educational programmes aimed at patients and their families. When possible, subgroup analyses could highlight variations in the effectiveness of interventions in different populations or hospital contexts, contributing to more precise and contextualised clinical recommendations.

In this sense, the findings of this review could have significant practical and political implications when translated into clear recommendations supported by performance indicators. The identification of cost-effective measures could inform policy decisions related to funding, continuing professional education and hospital accreditation processes. To ensure the effective dissemination and transfer of knowledge, the results will be communicated through scientific publications, technical summaries, and educational materials aimed at academic and clinical audiences, promoting their implementation at various levels of nursing practice and health policy formulation.

Finally, this research could reveal gaps in current practices, identify innovative strategies, and foster new lines of research, including the integration of digital technologies in infection prevention and the role of patient health education in reducing HAIs. In addition, it could stimulate economic evaluation studies on preventive technologies and implementation strategies. In summary, by offering a comprehensive and critical analysis of the existing evidence, this systematic review has the potential to enhance patient safety, strengthen infection prevention programmes, and promote a more responsible use of healthcare resources.

## Data Availability

No new data were created or analysed in this study. The original contributions presented in this study are included in the article. Further inquiries can be directed to the corresponding author.

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
