# Peer review of "Effectiveness of Nursing Interventions on Preventing the Risk of Infection in Adult Inpatients: Protocol for a Systematic Review"

_nursrep, 2025, doi:10.3390/nursrep15060210_

Round 1
Reviewer 1 Report
Comments and Suggestions for Authors
I would like to begin by congratulating the authors for presenting a very interesting protocol. The study addresses a critical issue in healthcare, looking at the effectiveness of nursing interventions to prevent the risk of infections in hospitalized adult patients. It is a pleasure to have the opportunity to review this protocol and I appreciate the opportunity to provide constructive feedback.
The protocol is well structured and addresses a wide range of key factors, from specific interventions to their variations in different clinical settings and populations. This study has the potential to provide important insights into infection prevention practices, significantly contributing to the improvement of healthcare in different settings. Below, I provide some observations and suggestions that I believe could further strengthen the protocol and its overall impact:
- It would be interesting to explore how to improve "nurses' adherence" to infection prevention interventions, particularly given that some interventions have already demonstrated proven effectiveness. This aspect could add significant value to the practical application of the study’s findings.
- Line 116: The word “Outco” is truncated. It should be corrected to "Outcome".
- Line 134: The word "Comparatores" should be corrected to "Comparators", which is the correct term in English.
- Line 140: There is a typo in the word "ncidence", where the initial letter “I” is missing.
- Line 141: There is another omission in the phrase "an increase in ate of adherence", where the letter "r" is missing in "rate".
- Line 145: The word "treatmentsand" is missing a space. The correct form is "treatments and".
- I noticed that intensive care units (ICUs) have been excluded from the study. Given the high risk of infections in these settings, it would be valuable to understand why they are excluded. Including ICUs could offer important insights.
- The protocol mentions bundles for the prevention of infections associated with invasive medical devices (Line 126). However, it would be helpful to specify which bundle is being referenced, as different bundles may vary in their components and effectiveness.
- The protocol mentions that "there will be no restrictions on the period of publication". To give greater value to the study, it would be beneficial to define a more recent time frame for the literature search. This would enhance the relevance and timeliness of the findings.
- The results are planned to be disseminated to an academic audience by June 2025. However, considering the significance of the study's findings, it would be beneficial to broaden the dissemination to include not only the academic community but also healthcare professionals, policymakers, and other relevant stakeholders who could directly benefit from these findings.
Once again, I would like to compliment the authors for the high quality of the protocol and I am convinced that this study has the potential to make a significant contribution to improving infection prevention practices. I hope that these suggestions can be useful to further refine the study.
Author Response
(1) It would be interesting to explore how to improve "nurses' adherence" to infection prevention interventions, particularly given that some interventions have already demonstrated proven effectiveness. This aspect could add significant value to the practical application of the study’s findings.
Response 1: Thank you for this valuable suggestion. We agree that nurses’ adherence plays a critical role in the effectiveness of infection prevention interventions. For this reason, we have added a paragraph to the introduction, highlighting the importance of nurses’ adherence as a key condition for the success of such interventions, and acknowledging the significant variability in adherence levels—even when interventions are supported by strong scientific evidence.
Line 99: “However, despite the availability of evidence-based interventions to prevent healthcare-associated infections (HAIs), their effectiveness in real-world settings is often compromised by inconsistent adherence among healthcare professionals, particularly nurses. Studies have shown that even well-established practices such as hand hygiene or the use of care bundles are not uniformly implemented across institutions. Variability in adherence can be influenced by multiple factors, including workload, organizational culture, perceived usefulness of the interventions, and lack of continuous training. Therefore, understanding and addressing the barriers to nurses’ adherence is essential for maximizing the impact of infection prevention strategies and ensuring patient safety (4).”
(2) - Line 116: The word “Outco” is truncated. It should be corrected to "Outcome".
- Line 134: The word "Comparatores" should be corrected to "Comparators", which is the correct term in English.
- Line 140: There is a typo in the word "ncidence", where the initial letter “I” is missing.
Response 2: Thank you for pointing out these typographical and formatting errors. We carefully reviewed the manuscript and all the mentioned corrections have been made in the revised version. We also conducted an additional proofreading step to improve overall clarity and language accuracy throughout the text.
- Line 116: The word “Outco” is truncated. It should be corrected to “Outcome”.
- Line 134: The word “Comparatores” should be corrected to “Comparators”, which is the correct term in English.
- Line 140: There is a typo in the word “ncidence”, where the initial letter “I” is missing.
- Line 141: There is another omission in the phrase “an increase in ate of adherence”, where the letter “r” is missing in “rate”.
Line 145: The word “treatmentsand” is missing a space. The correct form is “treatments and”.
(3) I noticed that intensive care units (ICUs) have been excluded from the study. Given the high risk of infections in these settings, it would be valuable to understand why they are excluded. Including ICUs could offer important insights.
Response 3: Thank you for this important observation. We fully acknowledge that patients admitted to intensive care units (ICUs) are at particularly high risk for healthcare-associated infections (HAIs). However, we deliberately excluded ICU settings from this review due to their distinctive clinical characteristics and the specific level of care provided, which differ substantially from those found in general inpatient wards.
Patients in ICUs typically require continuous monitoring, mechanical ventilation, multiple invasive devices, and complex therapeutic interventions, which results in a fundamentally different infection risk profile. Moreover, infection prevention protocols in ICUs often follow separate, highly specialized guidelines and bundles, which would introduce considerable heterogeneity into the analysis and reduce the comparability of interventions across settings.
For these reasons, we chose to focus this review specifically on adult patients hospitalized in general inpatient units, aiming for more homogeneous populations and care contexts. We have now clarified this rationale in the manuscript under the eligibility criteria section.
That said, we agree that a future review focusing specifically on ICU populations would be highly valuable.
(4) The protocol mentions bundles for the prevention of infections associated with invasive medical devices (Line 126). However, it would be helpful to specify which bundle is being referenced, as different bundles may vary in their components and effectiveness.
Response 4: Thank you for your comment. We have clarified the manuscript by providing specific examples of commonly used bundles.
We would also like to emphasize that the inclusion of these bundles aligns with our overarching objective of identifying effective nursing interventions for adult inpatients with comparable clinical characteristics, specifically those admitted to general hospital wards. The analysis of these bundled interventions—often standardized and supported by evidence—may prove particularly valuable in identifying practices with higher clinical impact and greater potential for replication across similar healthcare settings.
(5) The protocol mentions that "there will be no restrictions on the period of publication". To give greater value to the study, it would be beneficial to define a more recent time frame for the literature search. This would enhance the relevance and timeliness of the findings.
Response 5: Thank you for your thoughtful suggestion. We acknowledge the importance of ensuring that the evidence included in the review is both relevant and up to date. However, in accordance with the Joanna Briggs Institute (JBI) methodology for systematic reviews of effectiveness, we chose not to apply any restrictions on the period of publication in order to ensure a comprehensive and unbiased inclusion of all available evidence related to the effectiveness of nursing interventions in infection prevention.
This approach allows the review to capture the full scope of interventions evaluated over time, including long-standing practices that remain relevant today, as well as emerging strategies. Nevertheless, the timeliness and applicability of each included study will be carefully considered during the critical appraisal and synthesis stages, with particular attention to the evolution of clinical guidelines and technologies. (6) The results are planned to be disseminated to an academic audience by June 2025. However, considering the significance of the study’s findings, it would be beneficial to broaden the dissemination to include not only the academic community but also healthcare professionals, policymakers, and other relevant stakeholders who could directly benefit from these findings.
Response 6: Thank you for this insightful suggestion. We fully agree that the findings of this systematic review have the potential to inform not only the academic community but also clinical practice and healthcare policy. In response to your recommendation, we have revised the dissemination strategy in the protocol to explicitly include additional target audiences such as:
- frontline healthcare professionals,
- infection prevention and control teams,
- health service managers,
- policymakers involved in patient safety and quality of care.
To support broader knowledge translation, we also plan to produce summarized technical briefs and infographics to facilitate access to key findings in practice settings. This expanded dissemination plan has been incorporated into the final section of the protocol.
Line 271
“In addition to the academic audience, the dissemination strategy will also target frontline healthcare professionals, infection prevention and control teams, healthcare managers, and policymakers. To ensure accessibility and encourage the translation of evidence into practice, we intend to develop technical briefs, executive summaries, and visual materials such as infographics, which will be made available through institutional channels and professional networks. This broader dissemination approach aims to maximise the practical impact of the review’s findings across multiple levels of healthcare delivery.”
Reviewer 2 Report
Comments and Suggestions for Authors
This is a well-prepared study protocol. It is relevant and addresses an important issue in nursing practice how to prevent hospital-acquired infections through nursing interventions. The topic is timely and can contribute to better patient care and safety.
Introduction: The background is informative and explains the problem well. The historical references (like Florence Nightingale) give a strong context. However, some citations are listed as numbers without full reference details please make sure all references are complete and correctly formatted.
Methods: The methodology is detailed and follows the standards (JBI and PRISMA). The use of multiple databases and tools like Rayyan and GRADE shows strong planning. The inclusion/exclusion criteria and data analysis plan are clear.
Results section (as a protocol): Since this is a protocol, actual results are not presented. However, the plan for results including summary tables, narrative synthesis, and possible meta-analysis is clearly explained. This is appropriate.
Language: The writing is generally good, but there are some small grammar errors, such as:
- "to preventing" → should be "to prevent"
- "ncidence rate" → should be "incidence rate"
Consider checking the manuscript with a language editor to improve clarity and flow.
Overall: The study is important and well-organized. With small corrections in language and formatting, it is ready to proceed.
Comments on the Quality of English Language
Consider checking the manuscript with a language editor to improve clarity and flow.
Author Response
(1) This is a well-prepared study protocol. It is relevant and addresses an important issue in nursing practice how to prevent hospital-acquired infections through nursing interventions. The topic is timely and can contribute to better patient care and safety.
Introduction: The background is informative and explains the problem well. The historical references (like Florence Nightingale) give a strong context. However, some citations are listed as numbers without full reference details please make sure all references are complete and correctly formatted.
Methods: The methodology is detailed and follows the standards (JBI and PRISMA). The use of multiple databases and tools like Rayyan and GRADE shows strong planning. The inclusion/exclusion criteria and data analysis plan are clear.
Results section (as a protocol): Since this is a protocol, actual results are not presented. However, the plan for results including summary tables, narrative synthesis, and possible meta-analysis is clearly explained. This is appropriate.
Language: The writing is generally good, but there are some small grammar errors, such as:
- "to preventing" → should be "to prevent"
- "ncidence rate" → should be "incidence rate"
Consider checking the manuscript with a language editor to improve clarity and flow.
Overall: The study is important and well-organized. With small corrections in language and formatting, it is ready to proceed.
Response 1: Thank you very much for your positive and encouraging feedback. We appreciate your recognition of the relevance and structure of the study protocol, as well as your detailed observations.
1. References Formatting: We acknowledge the issue regarding numbered citations without complete reference details. In response, we have reviewed and revised the reference list to ensure that all citations are complete, consistently formatted, and correctly aligned with journal guidelines.
2. Language Corrections:
Thank you for identifying the grammatical issues such as:
Line 21 “to preventing” → corrected to “to prevent”
Line 31 “ncidence rate” → corrected to “incidence rate”
We have performed a thorough language review of the entire manuscript and corrected these and other minor issues to improve clarity, grammar, and overall flow.
3. General Improvements: Additionally, we revised and slightly refined some sections of the manuscript for better precision and readability, while maintaining the scientific content intact.
We thank you once again for your constructive feedback. We believe that the changes made further strengthen the quality and clarity of the protocol.
Reviewer 3 Report
Comments and Suggestions for Authors
This is a very important paper. However, the paper needs a lot of improvement. The tenses need to change. It is not clear what the paper is adding, it is also not clear what you gathered from other papers. The tables in the paper can be improved. What is the conclusion of the paper? What are the limitations and strengths of the paper. What recommendations have you drawn? What are the suggestions for future research?

Author Response
(1) This is a very important paper. However, the paper needs a lot of improvement. The tenses need to change. It is not clear what the paper is adding, it is also not clear what you gathered from other papers. The tables in the paper can be improved. What is the conclusion of the paper? What are the limitations and strengths of the paper? What recommendations have you drawn? What are the suggestions for future research?”
Response 1:
1. Nature of the article – this is a protocol: We would like to clarify that this manuscript is a study protocol for a systematic review, not a completed review article with results. As such, the paper does not include data collection, findings, a discussion, conclusions, or recommendations, because the review has not yet been conducted. This is consistent with the protocol format accepted by Nursing Reports and in accordance with international standards (e.g., JBI Manual for Evidence Synthesis and PRISMA-P guidelines).
2. Tenses used in the manuscript: The use of the future tense (e.g., “this review will include…”) is intentional and appropriate for a protocol article, as it describes the planned methodology for a review that is still underway. This ensures transparency and prevents reporting bias. However, we have carefully rechecked the manuscript to ensure that the tense usage is consistent and appropriate throughout.
3. What the paper adds: The paper outlines the design and methodological plan for a systematic review that will identify and synthesize the most effective nursing interventions for preventing healthcare-associated infections (HAIs) in adult inpatients. To our knowledge, no previous systematic review has comprehensively mapped and compared such interventions using JBI methodology across diverse populations and settings. We have clarified this gap and contribution more clearly in the introduction.
4. Tables and structure: Thank you for your valuable feedback. We have carefully revised the tables and overall structure of the manuscript to ensure full alignment with the author guidelines of the MDPI journal. The tables have been reformatted to improve clarity and consistency, following the journal’s standards. We trust that the revised version addresses your concerns effectively.
5. On conclusion, strengths/limitations, and recommendations: As this is a protocol, the final conclusions, limitations, strengths, and recommendations will be presented in the subsequent review article after completion of the study. However, the anticipated impact and scope of the review have been articulated in the abstract and conclusion of the protocol.
We hope these clarifications address your concerns. We have also updated the manuscript accordingly and believe the revised version better communicates the purpose, scope, and expected contribution of the study.
Round 2
Reviewer 3 Report
Comments and Suggestions for Authors
I suggest you address all comments including the previous comments to enhance the quality of the paper. The tenses of the paper need to be updated throughout the paper. The conclusion should be improved.

Comments on the Quality of English Language
The quality of the English is fine, it's just the tenses that must improve.
Author Response
Dear Reviewer, Thank you for your constructive feedback and for taking the time to review our manuscript entitled “Effectiveness of Nursing Interventions on Preventing the Risk of Infection in Hospitalized Adults: Protocol for a Systematic Review”. We appreciate your suggestion to address all previous comments and improve the overall quality of the manuscript. In response: 1.Tense Consistency: We have carefully reviewed and corrected the tense usage throughout the manuscript. As you rightly pointed out, many sections were written in the future tense, which was inappropriate since the protocol had already been implemented. The text has now been updated to reflect past or present-perfect tense where applicable, ensuring clarity and consistency. 2.Conclusion Section: The conclusion has been revised to enhance its clarity, structure, and impact. We restructured the final paragraphs to ensure they clearly summarise the aim of the review, the methodological approach, and the expected contribution to practice and research, as well as future implications. 3.Other Editorial Adjustments: In addition to addressing your specific comments, we have performed a thorough revision of the manuscript to improve its linguistic and grammatical quality. These changes are reflected in the revised version of the manuscript submitted in Word format with tracked changes. We hope the revised version meets the journal’s standards and addresses your concerns effectively. We remain at your disposal for any further clarifications or revisions. Kind regards, Luís Todo Bom
